# Lighting Strategies to Increase Nighttime Pedestrian Visibility at Midblock Crosswalks

**Rajaram Bhagavathula** * and **Ronald B. Gibbons**

Virginia Tech Transportation Institute, Virginia Tech, Blacksburg, VA 24061, USA
* Correspondence: rbhagavathula@vtti.vt.edu

**Abstract:** In the last decade, pedestrian fatalities at night, especially at midblock locations, have been increasing at an alarming rate. Lighting is an effective countermeasure in reducing nighttime crashes. However, few studies have evaluated the effects of crosswalk lighting on pedestrian visibility at midblock locations. There is an existing need to develop lighting designs that increase pedestrian visibility. Further, the safety effects of lighting have never been directly compared to other pedestrian-crossing treatments (such as flashing signs, rectangular rapid flashing beacons (RRFBs), etc.). Thus, in order to make effective recommendations for increasing nighttime pedestrian visibility, it is important to compare the visibility benefits of crosswalk lighting designs with and without pedestrian-crossing treatments. This study evaluated the visual performance of five midblock crosswalk lighting designs along with two pedestrian safety countermeasures at three light levels on a realistic midblock crosswalk. Visual performance was measured by calculating the distance at which the participants could detect a child-sized mannequin under the evaluated conditions. The results showed that midblock crosswalks should be illuminated to an average vertical illuminance of 10 lux to ensure optimal pedestrian visibility. Lighting designs that render the pedestrian in positive contrast (area in front of the crosswalk is illuminated) are recommended to increase pedestrian visibility. It is also recommended that pedestrian-crossing treatments, such as RRFBs and flashing signs, should be used with lighting to increase nighttime visibility.

**Keywords:** pedestrian visibility; lighting; crosswalk; midblock; rectangular rapid flashing beacons; flashing signs; pedestrian safety

## 1. Introduction

According to a Governors Highway Safety Association (GHSA) 2017 report, the number of pedestrian fatalities since 2007 has increased by 27%, while all other traffic deaths decreased by 14% [1]. Nationally, on average, about 75% of pedestrian fatalities occur after dark; in some states, the estimate is as high as 84% [1]. However, this statistic is even more severe when considering the fact that only about 25% of all traffic volume occurs after dark [2]. This means that during the time of day when the least number of vehicles are on the road, the greatest number of pedestrians are killed in crashes. Furthermore, as mentioned earlier, pedestrian deaths are the only category of traffic deaths that are increasing, while other categories of severe crashes are decreasing. This shows a heightened need to add or improve safety measures in order to protect areas of roadway traffic with high pedestrian volume, especially after dark.

One of the factors contributing to the pedestrian fatality rate is a lack of light. The International Commission on Illumination (CIE) states that the reason fatal road accident rates during darkness are so high is mainly due to reduced visibility [2]. Since approximately 90% of the information drivers use to navigate the roads is visual [3], avoiding pedestrians becomes more challenging with less light. In fact, fatal road crash rates during darkness are approximately three times greater than those during daylight [2]. Street

lighting increases visibility, augments vehicle headlamps, and provides more information about the surrounding area; consequently, it can lead to fewer crashes [4].

Very few studies have been conducted in the area of crosswalk lighting and pedestrian visibility. One of the earliest studies conducted on pedestrian visibility at intersection crosswalks was by Freedman and Janoff [5]; they reported that increasing the intensity of light resulted in an increase in the time available for drivers to respond and recommended an average horizontal illuminance of 75 lux for crosswalks. A before-and-after study conducted in Israel by Polus and Katz [6] reported that lighted crosswalks had significantly lower nighttime pedestrian crashes.

Pedestrian visibility studies conducted in Switzerland showed that rendering pedestrian in positive contrast (i.e., pedestrians are illuminated from the approach side, rendering them brighter than the background) reduced the pedestrian–vehicle crashes by two-thirds [7]. Pedestrians can be rendered in positive contrast by increasing the vertical illuminance on them from the vehicle approach direction to a sufficient level to overcome ambient light levels and vehicle headlamps from the opposing direction. The lighting design that rendered the pedestrians in positive contrast was compared to the existing design in a field test [8], which showed that the crosswalk lighting design that rendered the pedestrians in positive contrast provided significant benefits over the conventional one. The benefits of positive contrast on pedestrians was also reported in research conducted in realistic nighttime environments (visibility evaluations not conducted in laboratory settings or in computer simulations). Edwards and Gibbons [9] measured detection distances of pedestrians under different levels of vertical illuminance and reported that increasing the vertical illuminance on pedestrians increases the distance at which drivers can detect them.

All the of abovementioned research used fixed overhead lighting to illuminate pedestrian crosswalks; some of the recent research in pedestrian visibility used bollard type lights to illuminate pedestrian crosswalks. Bullough and Zhang [10] reported a study exploring different ways to illuminate crosswalks for potential improvements in motorist visibility of pedestrians and pedestrian safety. The study consisted of photometric simulations of various crosswalk lighting designs and in surveying individuals with expertise in fields of transportation, transit operations and public safety, specifically, in order to analyze the visual performance, glare and economic impacts of each lighting system. The responses concluded that the bollard-based lighting for crosswalks increased pedestrian lighting and reduced costs when compared to installing fixed overhead lighting. In a field test where four pedestrian crosswalk lighting configurations were evaluated along with a bollard lighting system [11], it was reported that the bollard-based system resulted in the shortest identification times of targets (adult- and child-sized black silhouettes). Bullough and Skinner [12] also reported demonstrations conducted at two crosswalks in Aspen, Colorado, and Schenectady, New York, over a two-night period. In the demonstrations, LED bollard-level lighting was installed to illuminate the studied crosswalks. The findings showed the subjective judgements to be consistently generally positive, concluding that the light levels needed for visibility can be achieved without excessive glare or other negative consequences through bollard-level lighting. These findings are yet to be replicated by objective measures of visual performance.

There are some differences in the light levels required for optimum pedestrian visibility in crosswalks, and these depend on the approach used for lighting of pedestrians in crosswalks. Edwards and Gibbons [9], who used conventional overhead lighting for illuminating crosswalks, reported that a vertical illuminance level of 20 lux at a height of 1.5 m (5 feet) from the road surface resulted in good driver visual performance at midblock crosswalks, whereas Bullough and Skinner [13], who used a bollard lighting system to illuminate a crosswalk, reported that a vertical illuminance of at least 10 lux on the pedestrian at a height of 0.9 m (3 feet) is required to increase contrast and, thereby, visibility.

It is important to note that pedestrian visibility in bollard-based lighting has never been directly compared to overhead lighting in realistic roadway conditions where the drivers approached the crosswalk at speed. Further, bollard-based lighting might increase

transient glare for drivers approaching the crosswalk; however, glare control could be improved through the use of louvers or baffles [10]. Another disadvantage of the bollard-based lighting is that it involves placing additional fixed objects adjacent to the roadway, which will increase the crash risk of errant vehicles.

There may be an interaction between roadway lighting and other treatments, such as pedestrian signals and signs, in that the effectiveness of one treatment could be supplemented by the addition of another treatment (such as lighting). As such, these treatments need to be considered in the evaluation.

The American Association of State Highway and Transportation Officials (ASSHTO) published their Highway Safety Manual (HSM), which suggests that installing flashing beacons at proper locations may significantly lower the vehicle–pedestrian crash rate. It is also important to note that the HSM also states that the overuse of flashing beacons may lessen their effectiveness [14]. Flashing beacons are most effective as a warning of unexpected or dangerous conditions not readily visible to drivers. There are many different types of flashing beacons. Two primary ones are rectangular rapid flashing beacons (RRFBs) and in-street pedestrian signs.

RRFBs are cheaper alternatives to traffic signals and are made of user-actuated LEDs that supplement warning signs at signalized intersections or midblock crosswalks. They can be activated by pedestrians manually through the push of a button, or passively via a pedestrian detection system. Once activated, the RRFBs use an irregular flash pattern and are more effective in increasing driver yield rates than the traditional overhead beacons. A study performed by the Oregon Department of Transportation replaced signs with RRFBs at three of their crosswalks on roads. The improved crosswalk resulted in an averaged 62% increase in yield rate. The conclusion of the study was that RRFBs should be considered for installation on high-speed roadways or intersections (speeds greater than 35 mi/h (56.32 km/h)) with a presence of pedestrians and bicyclists and a history of crashes or the potential for them [15]. Motorists' yielding rate increased in another before-and-after study, in a range from 35% to 80%, following the installation of RRFBs at an unsignalized intersection [16]. It is important to note, however, that RRFBs were found to be more effective at night than during the day [17].

In-street pedestrian signs are placed on the roadways' center line, lane line, or median of a crosswalk. FHWA found that in-street pedestrian-crossing signs are very effective at drawing motorists' attention to the presence of pedestrians; however, the countermeasure was found to be less effective after dark [18]. In a study performed at Michigan State University by Bennet et al., the installation of in-street signs produced 80% yielding, and the combination of the in-street signs with RRFBs produced 85% yielding [19]. A TCRP/NCHRP 2006 project similarly concluded that in-street pedestrian signs have relatively high driver yielding rates at unsignalized and midblock locations. The projects' findings averaged a driver yield rate of 87% on two-lane roads with speed limits of 25–30 mi/h (32.18–56.32 km/h) and determined the signs to be a highly cost-effective method for increased drivers' yielding rate at uncontrolled locations [20]. A case study performed in Las Vegas, Nevada, San Francisco, California, and Miami-Dade, Florida, reported increased drivers' yielding rates between 13% and 46% after the installation of in-street pedestrian signs. An important note from this study revealed that two of the studied crosswalk locations had no change in pedestrian–vehicle conflict [21]. More studies report similar results: in-street pedestrian signs are effective in increasing pedestrian safety. The level of effectiveness of this countermeasure varies with each study as the crosswalks' roadway traffic characteristics and other conditions differ at each observed location.

The safety effects of roadway lighting have never been directly compared to other pedestrian-crossing treatments, mentioned above, due to differences in the methods used to assess safety. Thus, there is an existing need to evaluate pedestrian-crossing treatments at crosswalks so that effective recommendations for increasing nighttime pedestrian safety on roadways can be put forward.

This study had two goals. First, develop guidelines for the lighting of crosswalks at midblock locations that increase pedestrian visibility. Second, to compare the effectiveness of the lighting with pedestrian safety countermeasures such as RRFBs, signs, etc., so that recommendations for increasing pedestrian visibility can be made. In the current study, visual performance of drivers was evaluated at five crosswalk lighting designs and two pedestrian-crossing treatments. Results from the study are intended to facilitate the development of midblock crosswalk lighting design guidelines.

## 2. Materials and Methods

A human factors evaluation was conducted on the Virginia Smart Road at night in relatively clear weather conditions (no rain/snow/fog). The Smart Road is a 2.2-mile-long (3.5 km), controlled-access research facility built to U.S. highway specifications. A realistic midblock crosswalk was simulated on the Smart Road. The results of these experiments are highly generalizable and readily applicable to real roads. A series of crosswalk lighting designs (each luminaire placement under three lighting levels) that are representative of currently used designs were developed and installed.

### 2.1. Participants

Twenty-four participants were recruited to participate in the study. Participant sample size was based on prior research which measured driver visual performance in similar situations [22–24]. Two participant age groups (18–35 years and 65+ years) were used. These two age ranges captured a wide range of driving experiences as well as a broad range of visual capabilities since human eyes undergo many physiological changes with age that result in several effects, such as a decrease in visual acuity, a decrease in contrast sensitivity, and an increase in dark adaptation time [25–29]. Each of these age groups was also gender balanced. All the participants had a valid US drivers' license and a corrected visual acuity of at least 20/40 (6/12). Visual acuity was measured with the early treatment diabetic retinopathy study chart using an illuminator cabinet. All experimental activities were approved by the Virginia Tech Institutional Research Board. Participants were paid $30 per hour for their participation in the study.

### 2.2. Experimental Design

The experimental design for the experiment is shown in Table 1.

### 2.3. Independent Variables—Crosswalk Lighting Designs

2.3.1. Overhead Lighting

For the midblock crosswalk, four designs were evaluated. All luminaires were of Type II and were a CCT of 4000 K. Type II luminaires have a light distribution with a preferred lateral width of 25 degrees and are intended for installations where the fixture is placed at the edge of a roadway. In the first design, the crosswalk was illuminated on the approach side (rendering the pedestrian at the entrance in positive contrast—Figure 1a). In the second crosswalk design (Figures 1b and 2b), the exit of the crosswalk was illuminated, rendering the pedestrian in negative contrast. The third and final design considered lighting both the approaches of a two-lane road, with one lane for each direction of travel (Figures 1c and 2a). It should be noted that the representations in Figure 1 are not photometrically accurate and are merely illustrations to show the location of the luminaires with respect to the crosswalk. Further, in realistic roadway conditions, contrast is never constant and changes as a function of the distance between the observer's vehicle and the pedestrian at the crosswalk [30]. Thus, it would be very difficult to establish a contrast metric for pedestrians at crosswalks. The designation of positive and negative contrast in this study was purely to determine the position of the light source with respect to the pedestrian at the crosswalk.

**Table 1.** Experimental design-independent variables for midblock crosswalk.

| Independent Variable | Levels | |
| --- | --- | --- |
| Light level (average vertical illuminance) | Low—2 lux<br>Medium—10 lux<br>High—20 lux | |
| Crosswalk lighting designs | Overhead Lighting | • Positive Contrast<br>• Negative Contrast<br>• Staggered (Positive contrast in both directions) |
| | Crosswalk Illuminators | • TAPCO Safewalk®Illuminator–CW Illuminator TAPCO<br>• Salex LED Flood light–CW Illuminator Salex |
| Pedestrian-crossing treatments | Rectangular Rapid Flashing Beacons (RRFBs)<br>Flashing Pedestrian Sign | |
| Mannequin Location | Entrance of the Crosswalk<br>Middle of the Crosswalk<br>Exit to the Crosswalk | |
| Participant age | Younger (18–35 years)<br>Older (65+ years) | |

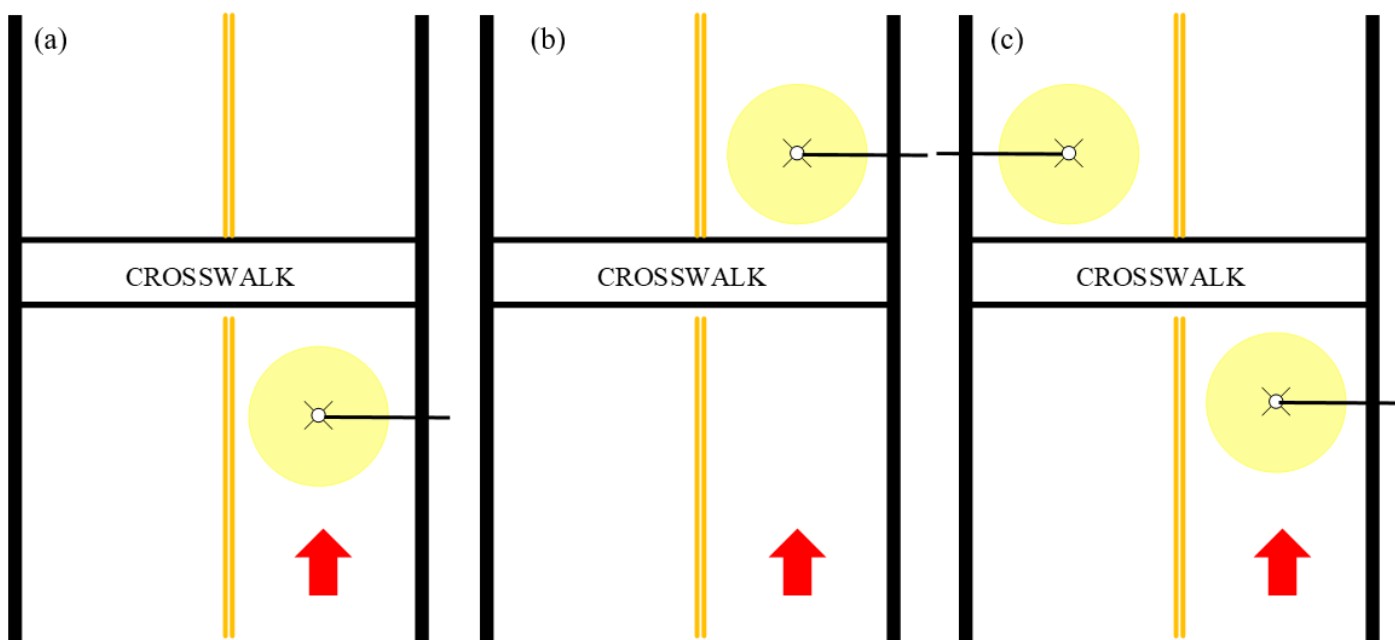

**Figure 1.** Midblock crosswalk lighting designs that were evaluated in the study: (**a**) positive contrast—luminaire was located before the crosswalk; (**b**) negative contrast—luminaire was located after the crosswalk; and (**c**) staggered—luminaires were located before the crosswalk in both directions of travel. Please note: figures are not photometrically accurate. The red arrows represent the direction of approach of a vehicle.

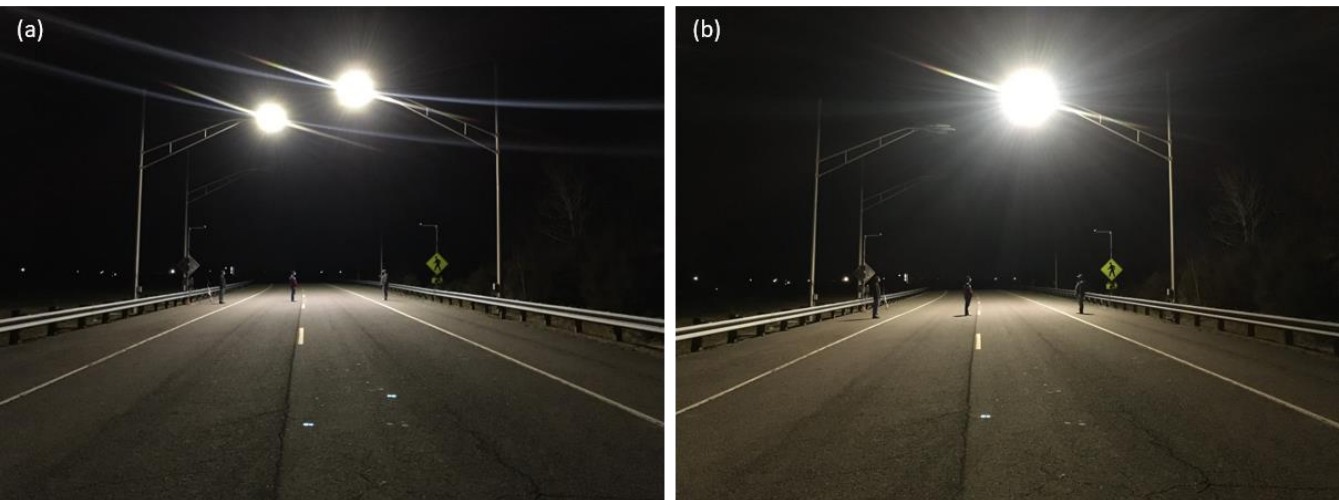

**Figure 2.** Midblock crosswalk lighting designs that were evaluated in the present study: (**a**) staggered—luminaires were located before the crosswalk in both directions of travel, and (**b**) negative contrast—luminaire was located after the crosswalk.

These designs also allowed for the testing of the effects of two types of contrast (positive and negative) on visual performance. Visual performance was tested under all these lighting designs to identify the best lighting design in terms of pedestrian safety at crosswalks. These luminaires were mounted at a height of 9.1 m (30 ft.) and were of 4000 K CCT.

### 2.3.2. Crosswalk Illuminators

In addition to the overhead lighting design for crosswalks, two commercial crosswalk illuminator products were also investigated. These two commercially available products used a narrow beam from LED flood lights mounted on poles (typically sign poles) adjacent to the roadway. The first one is TAPCO Safewalk® crosswalk illuminator (see Figure 3a) and the second one is Salex LED Floodlight crosswalk illuminator (see Figure 3b).

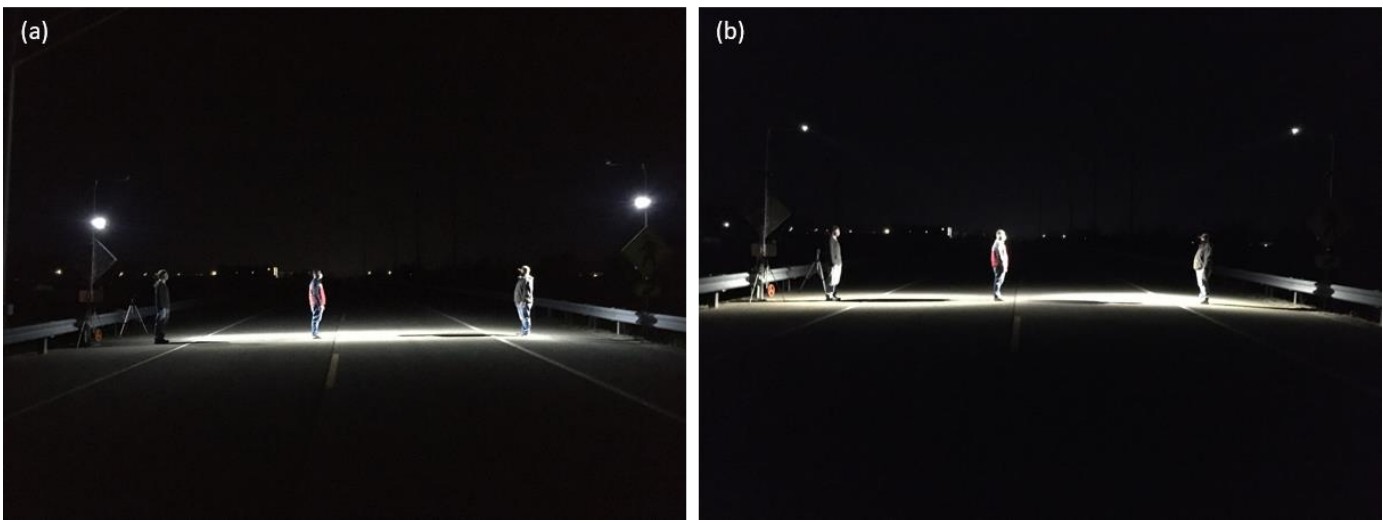

**Figure 3.** Photo. Crosswalk illuminators used in the midblock crosswalk lighting design, (**a**) TAPCO Safewalk® crosswalk illuminator and (**b**) Salex LED floodlight crosswalk illuminator.

### 2.4. Independent Variables—Crosswalk Light Levels

All the overhead crosswalk lighting designs were illuminated to three light levels based on the average vertical illuminance. Vertical illuminance was measured at a height of 1.5 m (5 ft.) from the surface of the roadway using a Minolta T10A illuminance meter. The low light level was established at 2 lux. The medium light level was established at 10 lux and it is based on the research of Bullough and Skinner [13]. The high light level was established at 20 lux based on the earlier research of Edwards and Gibbons [9]. The horizontal and vertical illuminance measurements of the crosswalk lighting designs are summarized in Tables 2 and 3.

**Table 2.** Horizontal illuminance measurements at the midblock crosswalk.

| Light Type | Light Level | Avg. (lux) | Min. (lux) | Max. (lux) |
|---|---|---|---|---|
| Positive and Negative Contrast | Low | 6.8 | 5.2 | 7.8 |
| Positive and Negative Contrast | Medium | 31.0 | 23.8 | 35.5 |
| Positive and Negative Contrast | High | 54.3 | 41.6 | 62.2 |
| Staggered | Low | 14.1 | 12.5 | 15.2 |
| Staggered | Medium | 57.8 | 51.3 | 62.1 |
| Staggered | High | 113.7 | 101.3 | 122.3 |
| CW Illuminator TAPCO | | 57.0 | 29.9 | 92.9 |
| CW Illuminator Salex | | 93.4 | 41.3 | 127.1 |

**Table 3.** Vertical illuminance measurements at the midblock crosswalk.

| Light Type | Light Level | Crosswalk Entrance (lux) | Crosswalk Middle (lux) | Crosswalk Exit (lux) | Avg. (lux) |
|---|---|---|---|---|---|
| Positive Contrast | Low | 2.2 | 2.5 | 1.6 | 2.1 |
| Positive Contrast | Medium | 10.2 | 10.7 | 8.6 | 9.9 |
| Positive Contrast | High | 19.8 | 19.0 | 12.6 | 17.1 |
| Negative Contrast | Low | 0.4 | 0.5 | 0.4 | 0.4 |
| Negative Contrast | Medium | 1.5 | 2.0 | 1.8 | 1.8 |
| Negative Contrast | High | 2.6 | 3.4 | 3.2 | 3.1 |
| Staggered Contrast | Low | 2.5 | 2.8 | 2.2 | 2.5 |
| Staggered Contrast | Medium | 10.4 | 11.7 | 8.1 | 10.1 |
| Staggered Contrast | High | 20.8 | 22.2 | 14.7 | 19.2 |
| CW Illuminator TAPCO | | 20.2 | 3.7 | 0.9 | 8.3 |
| CW Illuminator Salex | | 1.7 | 23.8 | 5.6 | 10.4 |

### 2.5. Independent Variables—Pedestrian-Crossing Treatments

Two pedestrian-crossing treatments were selected for evaluation at the crosswalk location. The first was a flashing pedestrian sign indicating the presence of a crosswalk. The second pedestrian-crossing treatment was the rectangular rapid flashing beacon (RRFB). Both of these treatments were shown to increase driver yield rates for pedestrians. RRFBs, in particular, have been shown to specifically be more effective at night [17]. The evaluation of pedestrian-crossing treatments was also conducted with and without the positive contrast lighting design mentioned above at the midblock crosswalk in order to understand their combined effect on driver visual performance.

### 2.6. Experimental Vehicles

Participants drove two identical instrumented vehicles (2016 Ford Explorers) equipped with data acquisition systems (DAS) that were connected to the vehicle's controller area network and on-board camera systems. The DAS collected kinematic data from the vehicle's CAN system, including vehicle speed, GPS coordinates, four video images (driver's face, forward roadway, left side of roadway, and right side of roadway), driver audio, and inputs from the experimenters.

## 2.7. Dependent Variables

Crosswalk lighting designs were assessed by measuring drivers' detection distance in a detection task. Detection distance is the distance at which pedestrians are visible and identifiable. Effective crosswalk lighting designs and pedestrian safety countermeasures increase detection distances.

## 2.8. Procedure

Driver visual performance was assessed under different crosswalk lighting designs. Participants were recruited for three sessions. In the first session, participants signed an informed consent document, and their visual acuities were checked to see if they met the requirements for the study. After the participant provided his or her consent, the in-vehicle experimenter escorted the participant to the experimental vehicle parked outside. The experimenter had the participant sit in the driver seat of the vehicle and demonstrated the seat and steering wheel adjustments. The experimenter then asked the participant to adjust as needed and to buckle their seat belt.

The in-vehicle experimenter entered the back seat of the vehicle and prepared the data collection equipment. The DAS recorded vehicle speed, GPS, and other network data from the vehicle, as well as video and audio inside and outside the vehicle. Once the DAS was ready, the experimenter instructed the participant to drive to the Smart Road. A speed limit of 35 mi/h (56 km/h) was established for the study and the in-vehicle experimenter kept track of the participants' speed during the experimental sessions.

The first lap of the first session was a practice lap. The practice lap was used to familiarize participants with where they would turn around, where the crosswalk was located, and to give them an opportunity to see the mannequins so they knew what to look for.

The experimental trials began once the participants indicated they were ready. Each time the participant drove through the test area, a different lighting design and light level would be presented at the midblock crosswalk. Additionally, the mannequins appeared at different locations relative to the crosswalks. As the participant drove, they were instructed to say the word "pedestrian," "kid," or "child" (whichever is easiest for them to remember) whenever they saw one of the child-sized mannequins. The in-vehicle experimenter pressed a handheld button each time a mannequin was identified. Later analysis of the data would determine the distance between this point and the mannequin, which was reported as the "detection distance." During pilot testing, full- vs. child-sized mannequins were evaluated for their suitability, as child-sized mannequins are smaller and more difficult (making them a more critical object) to detect than adult-sized ones. Each mannequin was 1.2 m (46 inches) in height. Child-sized mannequins were outfitted in gray-colored scrubs, as shown in Figure 4. Grey was chosen because it is a neutral color, and it is rendered similarly under different illuminance levels and lighting designs. Mannequins were located at the entrance, middle, and exit of the midblock crosswalk (see Figure 5).

This process of detections continued until the participant encountered all crosswalk lighting designs and light levels for each session. Presentation of crosswalk lighting designs and light levels were counterbalanced to reduce potential order-related confounding effects. The presentation of mannequins was randomized with "blanks" (i.e., no pedestrian presentation) to keep the participants from guessing and reduce learning-related confounding effects.

Once the experimental session was complete, the participants were asked to drive back to the Virginia Tech Transportation Institute building and were then dismissed. Participants drove 20 laps during each session. Each session lasted 1.5 to 2 h. Each participant took part in three experimental sessions, for a total participation time of approximately 6 h.

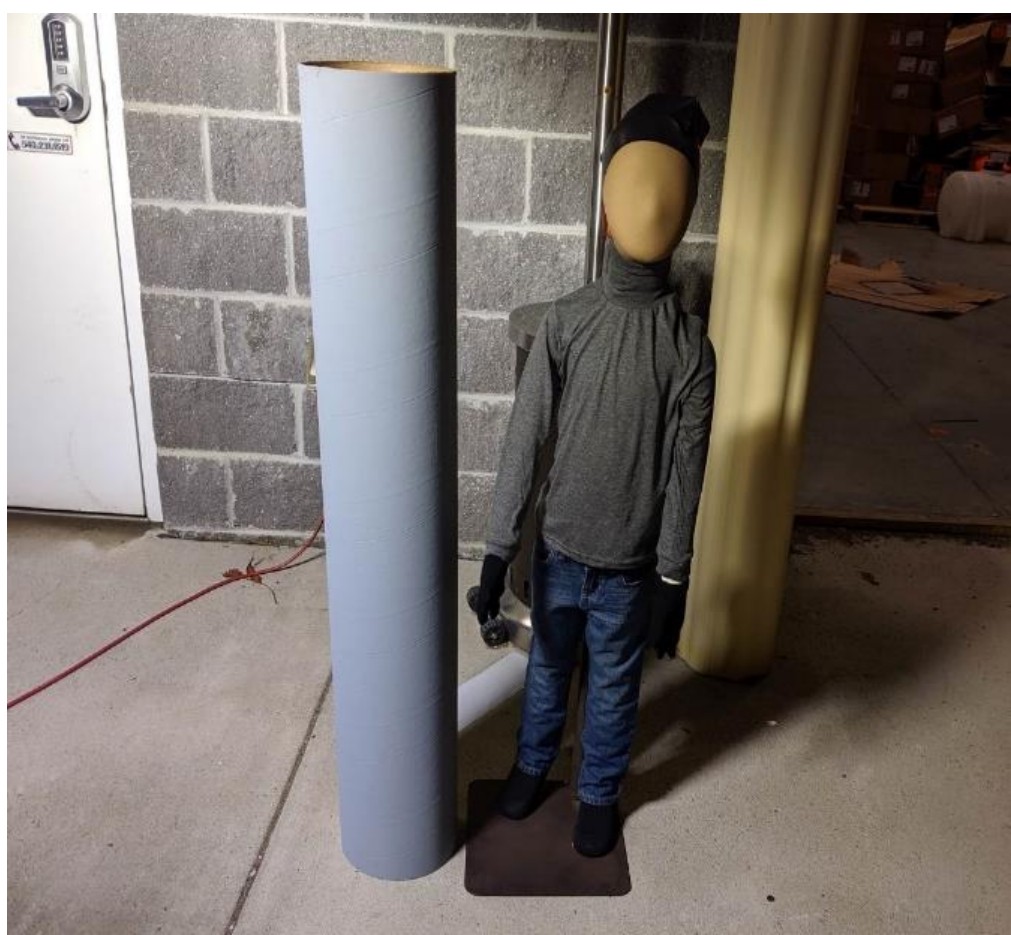

**Figure 4.** Child-sized mannequin wearing gray scrubs.

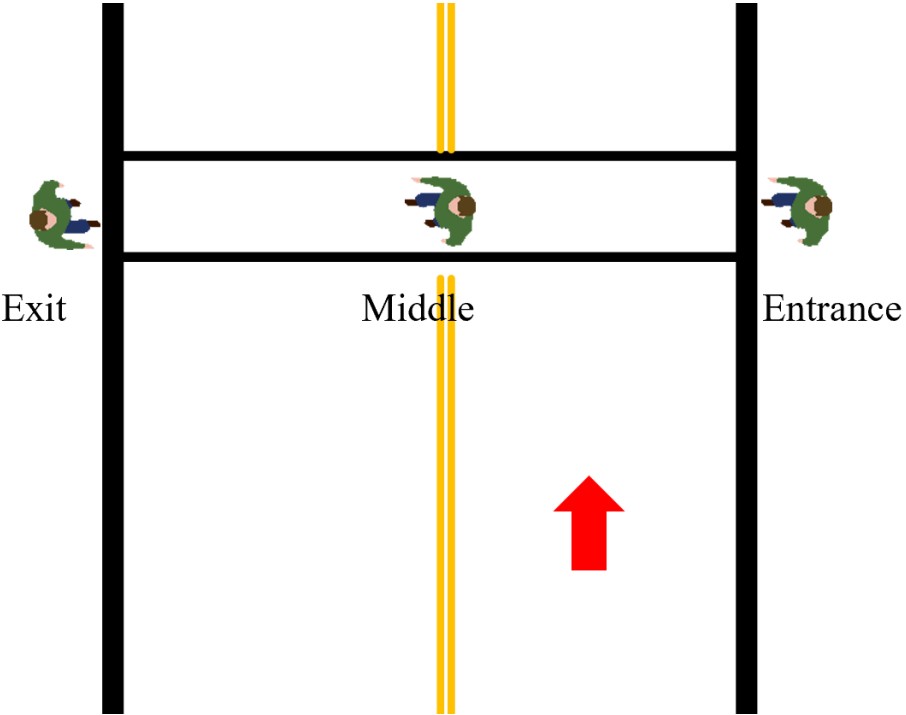

**Figure 5.** Mannequin locations used in the midblock detection task. The red arrow represents the direction of approach of a vehicle.

*2.9. Analysis*

A linear mixed model (LMM) was used to assess the effects of lighting design, lighting level, mannequin location, and age on detection distance. In order to facilitate the comparison across different crosswalk lighting designs, light levels, and pedestrian-crossing treatments, several independent variables were merged to form 19 discrete categorical levels of a single independent variable called "light condition". For example, the three overhead crosswalk lighting designs and three light levels were merged to give nine levels. The two pedestrian-crossing treatments by themselves and with a combination of an overhead crosswalk lighting design (positive contrast) at three light levels were merged to the give eight levels. Finally, the two crosswalk illuminators contributed to two levels. Overall, the "light condition" variable had 19 levels. Light condition, mannequin location, and age were included as fixed effects, and participant as a random effect in the LMM. The level of significance was established at $p < 0.05$. Where relevant, Tukey's honest significant difference was used for post hoc analyses. The analyses provided insights into the overall effects of crosswalk lighting design and light level on driver visual performance.

## 3. Results

The significant factors in the LMM are shown in Table 4. The main effects of age, light condition, and mannequin location were significant. The two-way interaction between age and light condition was also significant. Subsequent subsections provide additional details on the results regarding the light condition. Light condition refers to the combination of lighting design and light level at the midblock crosswalk.

**Table 4.** Significant statistical results from linear mixed model analysis of detection distance at the midblock crosswalk.

| Source | Num DF | Den DF | F Ratio | *p* Value |
|---|---|---|---|---|
| Age (A) | 1 | 23.00 | 19.09 | 0.0002 |
| Light Condition (LC) | 18 | 1401.20 | 83.55 | <0.0001 |
| Mannequin Location (ML) | 2 | 1401.00 | 29.69 | <0.0001 |
| A × LC | 18 | 1401.20 | 5.18 | <0.0001 |

The effects of overhead crosswalk lighting design and light level are shown in Figure 6. In all the lighting designs, an increase in light level resulted in an increase in the detection distances (see Figure 6). In all the lighting designs evaluated, detection distances in the medium and high light levels were significantly higher than those in the low light level. Further, there were no statistical differences between the medium and high light levels.

The detection differences between the lighting designs depended on the light level. When the overhead crosswalk lighting designs were at the low light level, the detection distances were significantly lower than the detection distances observed in both crosswalk illuminators (see Figure 7). Both pedestrian-crossing treatments, flashing sign and RRFB, had the shortest detection distances of all treatments tested. At the low light level, there were no differences between any of the overhead lighting designs (staggered, negative contrast, and positive contrast [with and without flashing sign and RRFB]). The detection distances in the staggered lighting design were only significantly higher than the flashing sign and RRFB.

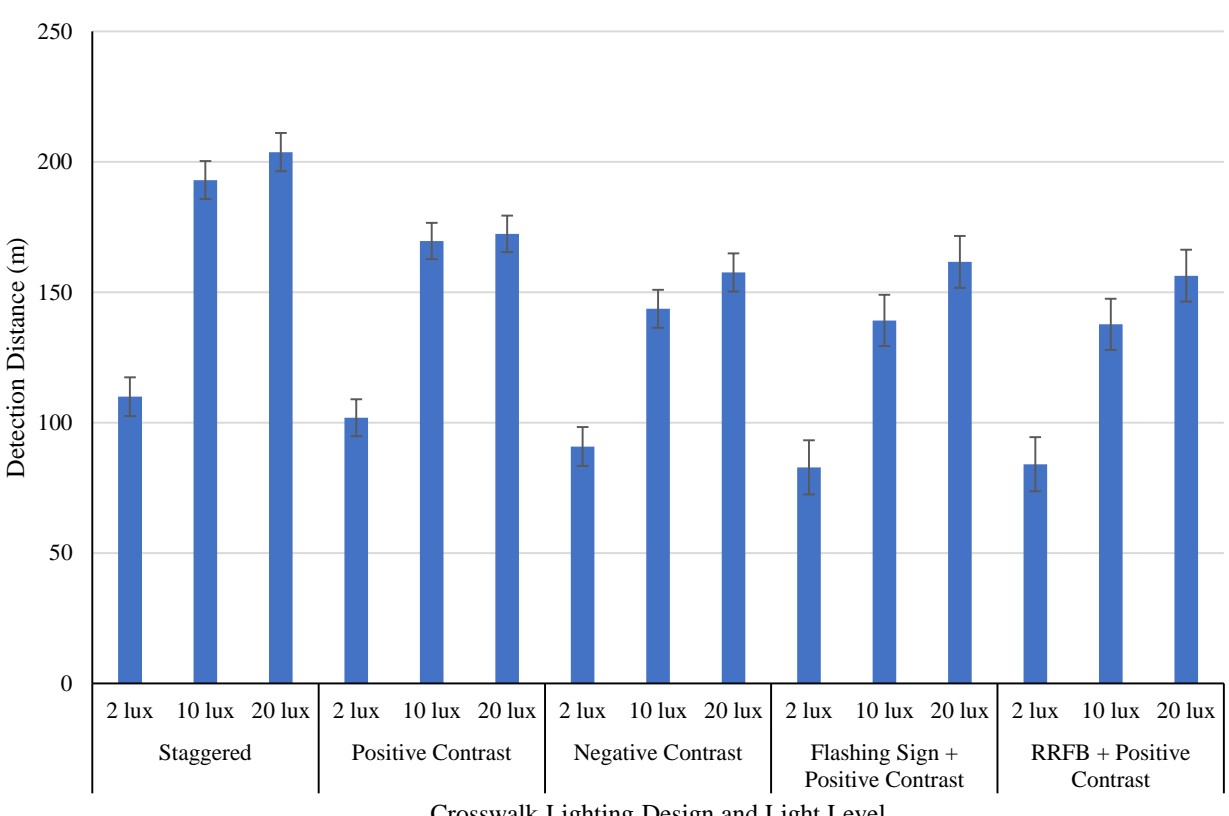

**Figure 6.** Effects of overhead lighting design and light level at the midblock crosswalk. Values are means of detection distances, and error bars indicate standard errors.

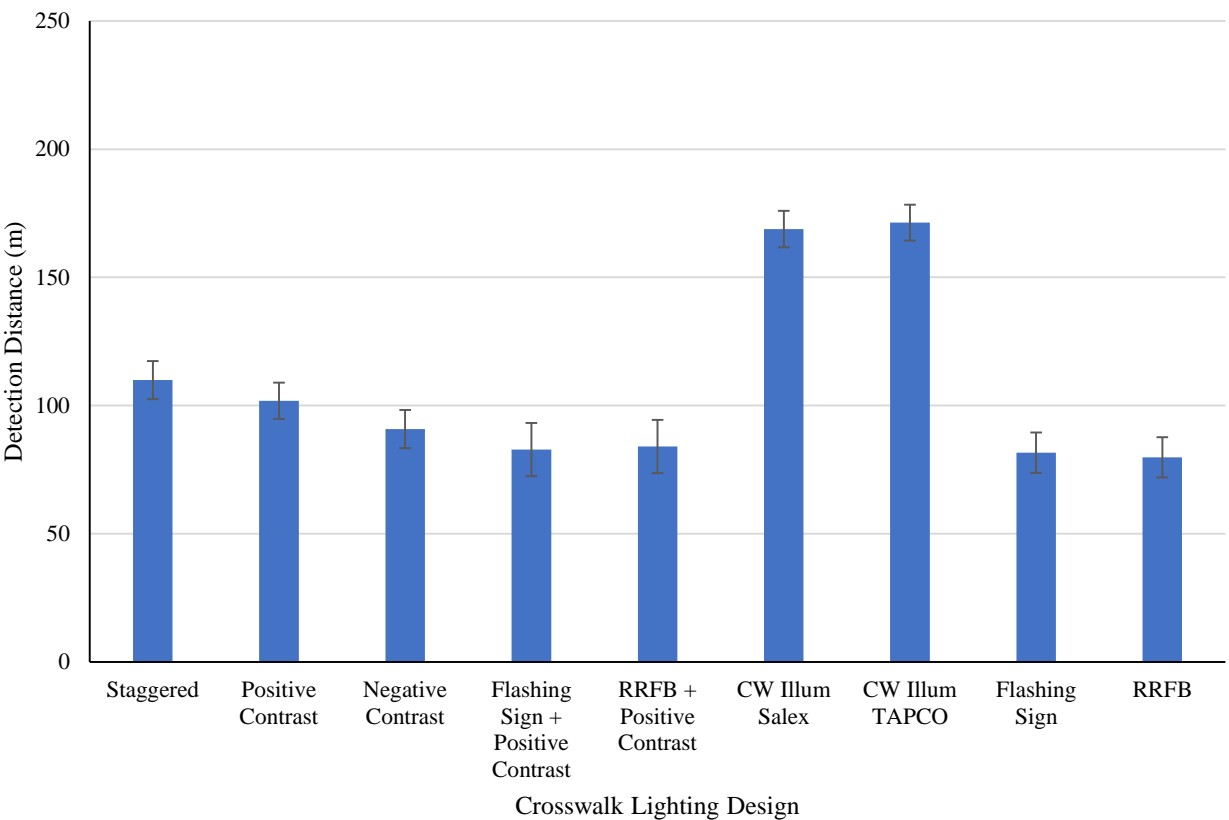

**Figure 7.** Effect of lighting design on detection distance at the low light level at the midblock crosswalk. Values are means of detection distances, and error bars indicate standard errors.

At the overhead lighting design's medium light level, the staggered, the positive contrast, and both crosswalk illuminators had the longest detection distances (see Figure 8). These were significantly higher than the rest of the lighting designs and the pedestrian-crossing treatments. Flashing sign and RRFB had the shortest detection distances among all lighting designs at the medium light level (Figure 8).

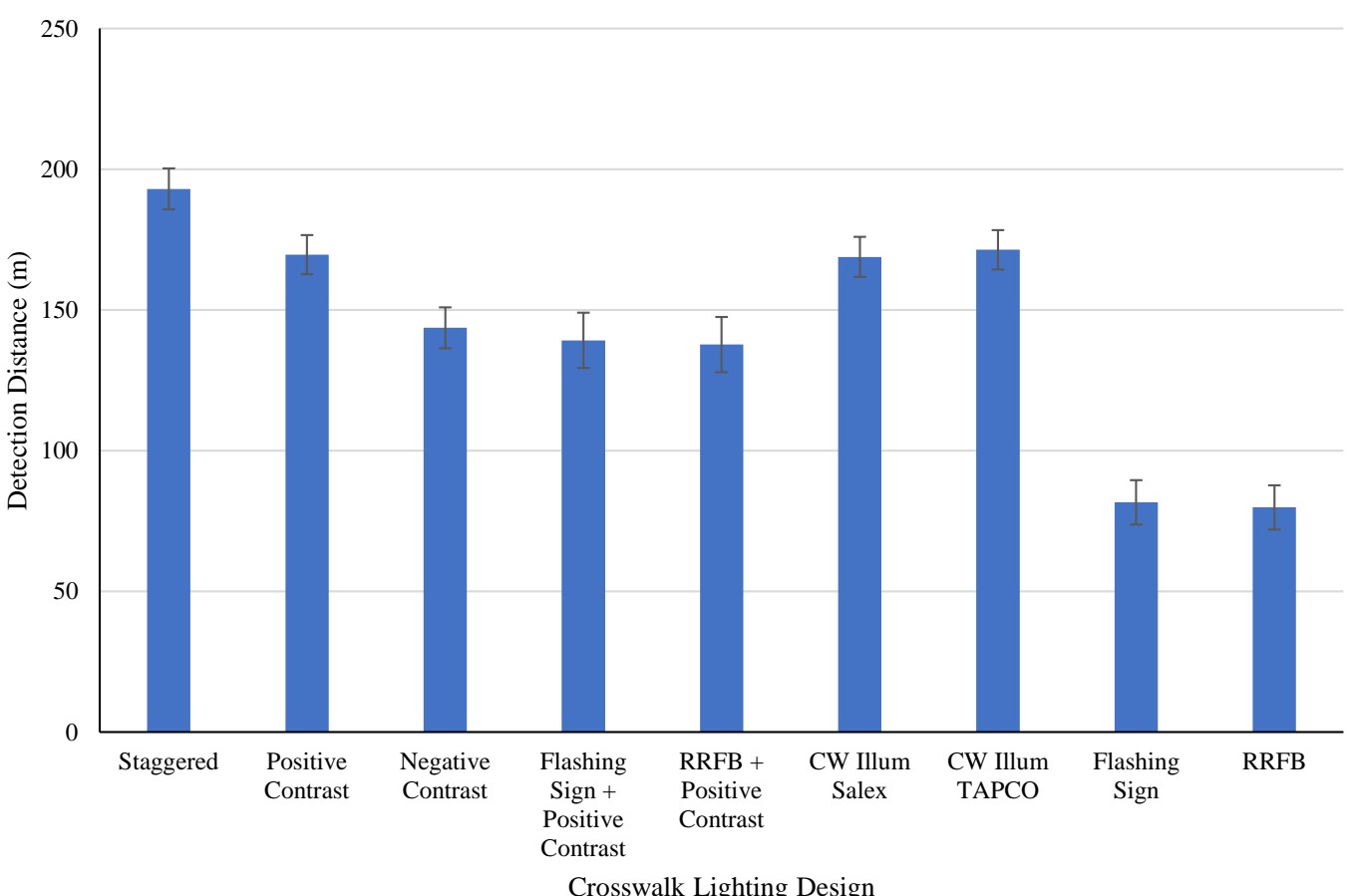

**Figure 8.** Effect of lighting design on detection distance at the medium light level at the midblock crosswalk. Values are means of detection distances, and error bars indicate standard errors.

At the overhead lighting design's high light level, the detection distances in the staggered lighting design were significantly longer than any of the other overhead lighting designs, pedestrian-crossing treatments, or a combination of them (see Figure 9). At the high light level, flashing sign and RRFB had the shortest detection distances (see Figure 9). There were no statistical differences between the negative contrast, positive contrast, positive contrast with flashing sign/RRFB, and both crosswalk illuminators.

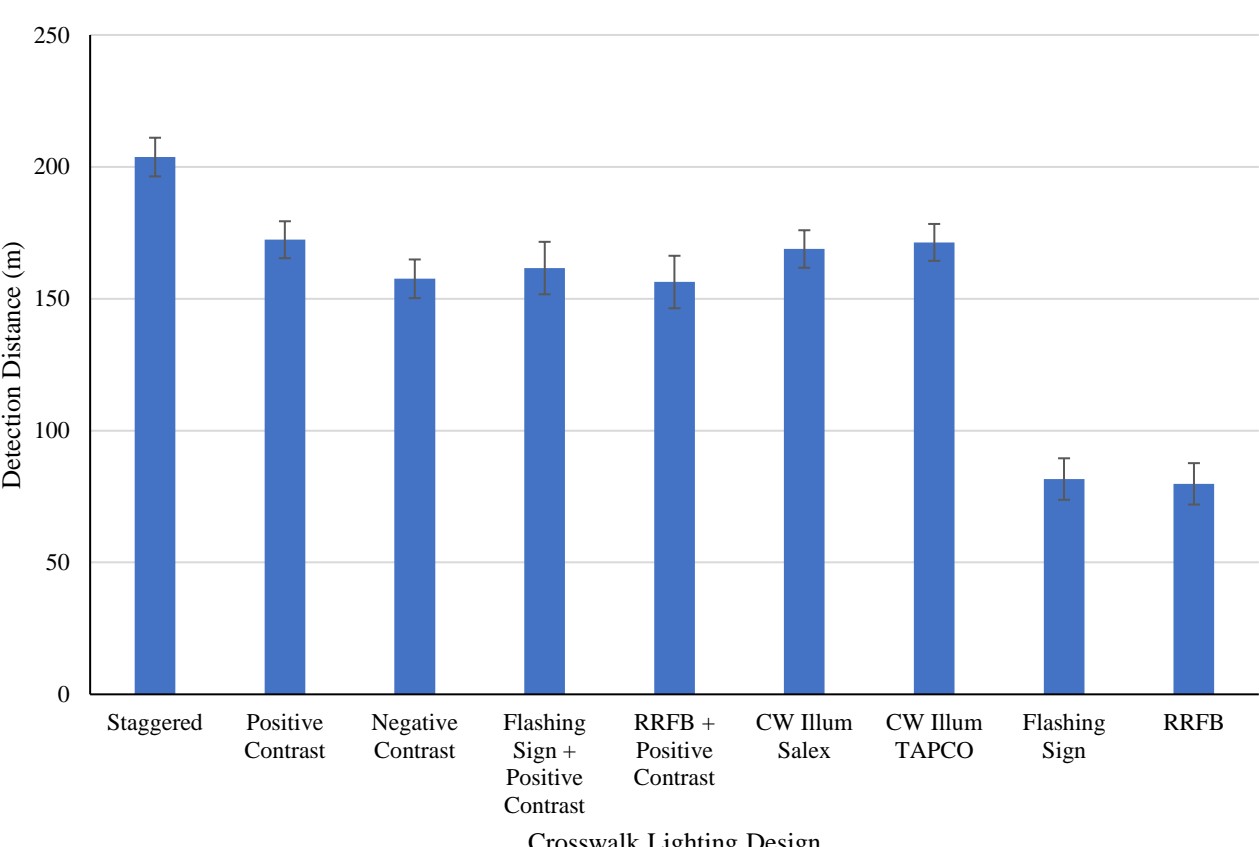

**Figure 9.** Effect of lighting design on detection distance at the high light level at the midblock crosswalk. Values are means of detection distances, and error bars indicate standard errors.

## 4. Discussion

This study had two goals. The first was to develop guidelines for the lighting of crosswalks at midblock locations that increase pedestrian visibility. The second goal was to compare the effectiveness of the lighting with pedestrian safety countermeasures, such as RRFBs, signs, etc., so that recommendations for increasing pedestrian visibility can be made. Two major findings are evident. First, there were visual performance differences between the tested lighting designs and light levels, with the staggered lighting design and crosswalk illuminators showing the best visual performance and pedestrian-crossing treatments (flashing sign and RRFB) the worst visual performance. Second, visual performance at midblock crosswalk plateaued at an average vertical illuminance of 10 lux.

Regarding the effects of crosswalk lighting design on driver visual performance, the results of the current study showed that the staggered lighting design had the best visual performance closely followed by the positive contrast lighting design and both crosswalk illuminators (TAPCO and Salex). The better visual performance of the staggered lighting design among all the overhead lighting designs (staggered, positive contrast [with and without RRFB and flashing sign], and negative contrast) could be attributed to the staggered lighting design rendering the pedestrian in higher positive contrast as a result of higher vertical illuminance (see Table 3). Among the overhead lighting designs, negative contrast lighting design had the lowest detection distance, and was almost on par with the detection distances that were observed for the pedestrian-crossing treatment combined with the positive contrast lighting design.

Within the overhead lighting designs, visual performance depended on the light level. An increase in the light level resulted in an increase in the detection distance. This result is consistent with existing research which looked at the effects of increasing light levels on pedestrian visibility [31,32]. However, beyond the medium lighting (average vertical illuminance of 10 lux), the visual performance plateaued, as evidenced by the lack of statistical

differences between the detection distances across medium and high light levels (vertical illuminance of 20 lux). The results from the current study also showed that visual performance in the staggered and positive contrast lighting designs was better than the visual performance in the negative contrast or the positive contrast with RRFB/flashing sign at light levels greater than 2 lux. These results reinforce results of earlier studies [9,33], which argued that rendering a pedestrian in positive contrast will increase their visibility. Positive contrast lighting designs could aid in pedestrian visibility as these could provide vital clues about the direction of the pedestrians' travel by discerning features such as hands, face, etc., unlike negative contrast, which only renders the pedestrians as silhouettes, making the feature discrimination difficult [30,34]. These results indicate that a vertical illuminance of at least 10 lux on the pedestrian at the entrance of the crosswalk when the luminaire is located in front of the crosswalk will increase pedestrians' visibility to approaching drivers. The vertical illuminance of 10 lux for pedestrian visibility is also supported by existing research which used bollard-based luminaires to illuminate pedestrians in a crosswalk [13].

Both crosswalk illuminators used in the current study had detection distances that were comparable to the best overhead crosswalk lighting designs (staggered and positive contrast) at a vertical illuminance of 10 lux and beyond. The longer detection distances under the crosswalk illuminators could be attributed to the higher vertical illuminance and, as a result, higher positive contrast for the pedestrian on a driver's approach to the crosswalk. These results could also indicate that additional illuminance provided by the overhead roadway lighting designs, specifically the staggered lighting design, increases the amount of visual information gathered by the drivers' eyes, thereby increasing visual performance. These results also demonstrate the benefits of illuminating the surrounding area adjacent to the roadway in order to increase drivers' visual performance at night. However, such designs should be carefully implemented, and adequate care should be taken to reduce the impacts on the surrounding ecology and environment.

The pedestrian-crossing treatments, RRFB and flashing signs, had the worst visual performance, as evidenced by the shortest detection distances among all the tested crosswalk lighting designs. Both pedestrian-crossing treatments do not illuminate the pedestrian in any way. The results showed that a combination of overhead lighting (positive contrast) and pedestrian-crossing treatments greatly increased the visual performance, as evidenced by the longer detection distances. The flashing lights seemed to draw attention away from the crosswalk even when both pedestrian-crossing treatments are used in conjunction with overhead lighting, as those combinations had lower detection distances than the staggered or positive contrast lighting designs. Based on the results of the current study, pedestrian-crossing treatments should be accompanied by lighting at crosswalks to increase pedestrian visibility at night. However, it is important to note that in the current study, the pedestrian-crossing treatments were evaluated only from a pedestrian visibility point of view at night; their effectiveness during the day in affecting driver yielding behavior has been clearly established by existing research [19–21].

This study had some limitations. First, this study only addressed visual performance from a drivers' point of view and does not account for pedestrians' visual performance or perceptions. This was carried out in order to simplify the experimental design and reduce exposure during the COVID-19 pandemic. Second, the child-sized mannequins, which were used as simulated pedestrians, did not move and were stationary. This simplification was made to ensure that the drivers could maintain the speed limit established for the study. Future research should aim to address these limitations.

## 5. Conclusions

Pedestrian visibility at midblock crosswalks is influenced by the lighting design, light level, and presence of pedestrian-crossing treatments. Based on the results of the study, the following recommendations can be made. Midblock crosswalks should be illuminated to an average vertical illuminance of 10 lux to ensure optimal visibility of pedestrians. Where overhead lighting is available, midblock crosswalk lighting designs that render the

pedestrian in positive contrast are recommended to increase pedestrian visibility. Luminaires located in front of the crosswalk will ensure that pedestrians are rendered in positive contrast. Where overhead lighting is not available, crosswalk illuminators can be used to illuminate midblock crosswalk locations. At night, it is recommended that pedestrian-crossing treatments, such as rectangular rapid flashing beacons and flashing signs, should be used in conjunction with overhead or crosswalk illuminators at the established vertical illuminance to ensure optimal pedestrian visibility at midblock crosswalks.

**Author Contributions:** Study conception and design: R.B.; data collection: R.B.; analysis and interpretation of results: R.B. and R.B.G.; draft manuscript preparation: R.B. All authors have read and agreed to the published version of the manuscript.

**Funding:** This research was funded by the Illinois Center for Transportation through grant number R27-202.

**Institutional Review Board Statement:** The study was conducted in accordance with the Declaration of Helsinki and approved by the Institutional Review Board of the Virginia Polytechnic Institute and State University. The Virginia Tech IRB reference number is 20-332, approved on 7 July 2020.

**Informed Consent Statement:** Informed consent was obtained from all subjects involved in the study.

**Data Availability Statement:** The data are not currently publicly available.

**Acknowledgments:** This research is based on the results of ICT-R27-202: Roadway Lighting's Effect on Pedestrian Safety at Intersection and Midblock Crosswalks. ICT-R27-202 was conducted in cooperation with the Illinois Center for Transportation, the Illinois Department of Transportation, and the U.S. Department of Transportation, Federal Highway Administration. This research also received luminaires and crosswalk illuminators which served as donations in-kind from LED Green Light International, Salex, and TAPCO. TAPCO also donated the flashing sign and the RRFB used in the study. The authors would also like to acknowledge the Virginia Tech's Open Access Subvention Fund for providing financial support in getting this article published.

**Conflicts of Interest:** The authors declare no conflict of interest.

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
