# Peer review of "Lighting Strategies to Increase Nighttime Pedestrian Visibility at Midblock Crosswalks"

_sustainability, doi:10.3390/su15021455_

Round 1

Reviewer 1 Report

The authors evaluated the visual performance of five mid-block crosswalk lighting designs and two pedestrian safety countermeasures at three light levels on a realistic midblock crosswalk simulated on the Virginia Smart Road. 24 participants divided into two gender-balanced age groups (18-35 years and 65+ years) drove 20 laps in three sessions during which the detection distance of the mannequins placed at the crosswalk was observed. Statistical analysis was performed over 4 groups of independent variables (average vertical illuminance, lighting designs, crossing treatments, and participant age), and drivers’ detection distance as the dependent variable. A linear mixed model (LMM) was used to assess the effects of lighting design, lighting level, and age on detection distance. Based on the study's results, they gave five recommendations for the design of the independent variables observed.

The abstract and the manuscript itself are clear, relevant to the field, and presented in a well-structured manner.

The references authors use in the introduction and discussion are older, but this only emphasizes the need for the presented study.

Their experimental design is appropriate for testing and comparison of the effectiveness of the lighting of crosswalks at mid-block locations with pedestrian safety countermeasures. The manuscript’s results are reproducible based on the details given in the methods section.

Figures, tables, images, and schemes show the data correctly, they are pretty easy to interpret and understand.

The authors' conclusions are consistent with the evidence and arguments presented.

I do have a few specific comments:

·         Please use the SI Units more consistently (please add the appropriate value in SI unit in brackets along with Imperial one in lines 124, 138, and 161)

·         Line 49 – please correct the 'increased'

·         Line 188 – please correct 1.d to 1.c.

Author Response

  • Please use the SI Units more consistently (please add the appropriate value in SI unit in brackets along with Imperial one in lines 124, 138, and 161)
  • Response: Units have been changed as suggested.
  • Line 49 – please correct the 'increased'
  • Response: Changed as suggested.
  • Line 188 – please correct 1.d to 1.c.
  • Response: Changed as suggested.

Reviewer 2 Report

The present study deals with the safety of pedestrians in the dark and investigating the effect of lighting. Undoubtedly, it is an interesting and important topic, but the authors must give detailed answers to the following concerns so that their article will be re-evaluated.

The objective of the paper is not clear in the abstract. The authors said that there is an existing need to develop guidelines for midblock crosswalk lighting. Do they want to develop a guideline? I believe no; they may provide some suggestions but not a guideline; as they said later that there is a need to compare the performance of pedestrian crossing treatments at crosswalks with and without lighting. So, it seems they want to conduct a before-after study (?) to provide some effective recommendations. Nevertheless, the authors must distinguish between the objective of the article and its application.

Following my previous comment, the methodology used in this study should be stated in the abstract. The authors should not expect their readers to guess what they have done.

The contribution of the paper was well discussed in the last two paragraphs of the introduction section. However, in my opinion, it should be reflected in the abstract in a few short sentences as well. I recommend removing unnecessary and repetitive content and explaining the purpose, method and contribution directly and briefly.

I do not understand the reason for using this number of participants and their age division. How can the authors of this article be sure that their sampling can include the entire community of drivers? Especially since they claim they want to have recommendations for guideline development.

How have the authors handled unobserved heterogeneity among drivers?

By repeating the experiment for different scenarios, how did you ensure that the previous knowledge (from the previous experiment) did not affect the drivers' behavior?

In the middle of the article, I fully realized what exactly you did and, for example, you did not conduct a common before-after study of crashes; but I deliberately did not change my first comment because it shows why I said that the methodology should be stated clearly in the abstract.

Figures (e.g., 6 and 7) show a lot of information. However, the reason for the various differences was not fully discussed. A strong paper does not just report the findings and discusses possible reasons for what was seen. I read the discussion section, but it does not discuss and compare all the findings shown in the figures.

Reviewer 3 Report

The paper „Lighting Strategies to Increase Nighttime Pedestrian Visibility at Midblock Crosswalks” deals with the crucial issue of adequate illumination of crosswalks. Nowadays, this problem is very often commented on in the literature. Generally, the layout of this paper seems to be proper, and the text is written well. The introduction, methods, description of the experiment performed, and discussion are clear and transparent. However, the main drawbacks of this study are connected with a rather substantive issue.

- First, it needs to be clarified at what height this 10 lux of vertical illuminance was achieved. Was it uniform in the entire area of the crosswalk? It is not possible. What equipment was used to measure it, and what was the measurement uncertainty?  

- The most important is the issue of contrast. The Authors try to analyze the visibility of pedestrians in positive and negative contrast. But it needs to be clarified what the exact value of this parameter was. There is a possibility of determining the contrast, but the knowledge of ambient brightness (e.g., the luminance of surroundings) is required. In real cases, the ambient brightness can be different for different locations. So, the recommendation to illuminate every crosswalk with an average vertical illuminance of 10 lux seems too strict.

Other issues that must be improved: 

- L.182 Please explain what „type II” means.

- Figure 1: Please provide a better technical drawing of analyzed cases (dimensions) and luminous intensity distribution curves of luminaires used as a separate figure.

- Table 4:  Please explain all the metrics used in this table.

- L.202 Please use the international system of units in the whole text same as in L.258

- The literature contains only 22 items. Many papers deal with this topic, and not all essential sources were cited. So, the literature should be extended.

The remarks mentioned above have to be explained and improved. I encourage the Authors to prepare the revised version of this paper. 

Round 2

Reviewer 2 Report

No further comment

Reviewer 3 Report

The authors took into account most of my comments and explain them in their answer. The article has been improved accordingly. Thank you very much.